# *Cnaphalocrocis medinalis* Moths Decide to Migrate when Suffering Nutrient Shortage on the First Day after Emergence

**DOI:** 10.3390/insects10100364

**Published:** 2019-10-20

**Authors:** Jia-Wen Guo, Ping Li, Jie Zhang, Xiang-Dong Liu, Bao-Ping Zhai, Gao Hu

**Affiliations:** Key Laboratory of Integrated Management of Crop Diseases and Pests (Ministry of Education), College of Plant Protection, Nanjing Agricultural University, Nanjing 210095, China; 2016202022@njau.edu.cn (J.-W.G.); 2017102102@njau.edu.cn (P.L.); 2014202037@njau.edu.cn (J.Z.); liuxd@njau.edu.cn (X.-D.L.)

**Keywords:** adult starvation, energy trade-off, reproduction, migration, triglyceride

## Abstract

Migration is a costly strategy in terms of reproduction output. Competition for limited internal resources leads to physiological management of migration-reproduction trade-offs in energy allocation. Migratory insects must choose to determine to allocate energy into reproduction or migration when confronted insufficient energy supply. Although nutrient shortage is known to stimulate insect migration to escape deteriorating habitat, little is known about when and how migratory insects make decisions when confronted by a nutritional shortage. Here *Cnaphalocrocis medinalis* (Lepidoptera: Pyralidae), a migratory rice pest in eastern Asia, was used to study the effects of starvation on reproductive traits, behavioral traits and energy allocation. The result showed that one or two days’ starvation before preoviposition did not significantly reduce the fertility (total egg per female laid) and flight capability (flight duration and distance) of both sexes *C. medinalis*. The preoviposition period was extended significantly only if moths were starved starting on the first day after emergence. Also, take-off percentage of moths starved since their first day increased significantly, and continued to increase even if supplemental nutrients were supplied as honey solution in later days. Moths starved on the first day appeared to allocate or transfer triglycerides into the thorax to maintain the migration process: the quantity of thoracic triglycerides did not differ with age, but abdominal triglycerides decreased with age if starvation continued. These results indicate that the first day post-emergence is a critical period for *C. medinalis* to decide to migrate or not in response to lack of food. This furthers our understanding of the population dynamics of migratory insects under natural conditions.

## 1. Introduction

Migration is an adaptive mechanism for insects to avoid adverse conditions and exploit temporary or patchy habitats [1,2]. In most insects, migration usually occurs in the pre-reproduction period [3], a phenomenon termed by Johnson [4] as “oogenesis-flight syndrome”. Many insects exhibit this behavior, such as *Myzus persicae* [5], *Aphis gossypii* [6], and *Gryllus firmus* [7]. There is a trade-off between migration and reproduction in these insects. In other words, the greater the amount of energy consumed for flight, the less remains for development and reproduction. A longer preoviposition period, lower fertility, lower reproductive energy investment, higher take-off percentage and stronger flight capacity are usually associated with migratory behavior of insects [8,9,10,11].

Food conditions have important effects on the reproduction and migration of insects [2,4,12,13]. In addition to the quality and quantity of food intake, the time of nutrient intake plays an important role in determining reproductive and migration behavior [14,15,16]. In nature, a majority of adult insects, such as *Lepidoptera*, *Diptera*, and *Hymenoptera* species, need nutrient supplements to enhance reproduction [17,18,19,20]. Adult nutritional status affects the reproductive behavior of insects, including mating [21,22] and oviposition [23,24,25]. In addition, since migration of most insects usually occurs when the adult is sexually immature [3], the decision by adult to migrate may be more directly affected by nutrient conditions in the adult stage than in the larval stage. Previous studies have suggested that the supplementation of nutrition in the adult affects insect migratory behavior [2,4]. For example, the migratory movement of *Danaus plexippus* depends on availability of nectar resources [26], and large numbers of these butterflies are found at locations where nectar is abundant during migration season [27]. Lack of supplemental nutrition in adults can promote insect migration, as demonstrated in *Bruchophagus roddi* [28], *Oncopeltus fasciatus* [29] and *Dysdercus cingulatus* [30]. However, little is known about how newly emerged migratory insects make decisions on allocating resources towards reproduction and migration when faced with nutritional deficiencies.

*Cnaphalocrocis medinalis* (Lepidoptera: Pyralidae) is an important long-distance migratory pest, that has had serious outbreaks in many Asian countries in recent decades, especially in China [31,32]. It cannot overwinter in temperate climates. In China, only a few individuals can survive during winter in areas of Southern Yunnan, Southern Guangxi, Southern Guangdong and Hainan [31,32,33]. Instead, infestations in temperate zones are initiated by windborne spring/summer migrants from the southern areas. A northward migration begins in March every year, and migrant progeny further expand population distributions to cover the rice-growing regions in China, Japan and the Korean Peninsula [31,32,33,34]. From September onwards, the general direction of *C. medinalis* becomes predominantly south bound [31,32,33,34].

Like most nocturnal migratory moths, *C. medinalis* perform ‘multi-stop’ migration in 3–5 consecutive nights, in which moths only take off at dusk, terminate migratory flight the following dawn, and then take-off again at the next dusk [33]. Adult *C. medinalis* exhibit “oogenesis-flight syndrome” during migration, which means migration occurs during the immature stage of reproductive development [33,34,35,36,37]. Therefore, the migratory *C. medinalis* generally has a long reproductive preparation stage (longer preoviposition period, lower mating percentage and less fecundity for female) and a strong migratory propensity (higher take-off percentage and strong flight capability in both sexes) [12,34,35,36,37] than a non-migratory moth. *C. medinalis* migrants also reserve sufficient energy for migration [35,36]. Lipids, stored primarily as triglycerides, are the main energy sources for migration in many nocturnal moths [9], and *C. medinalis* migrants have more lipids stored in thorax than residents [35,36]. After one or two days post-emergence, *C. medinalis* needs to acquire supplemental nutrients, which are often sucked from plant flowers in the field such as *Paspalum conjugatum*, *Amaranthus viridis*, *Ligustrum lucidum*, and *Gossypium* spp. [12]. Nutritional supplementation in the adult stage is especially important for *C. medinalis*. Lack of supplementary nutrition prolongs the preoviposition period, and has a significant negative impact on reproduction and longevity [12]. The level of ovarian development of *C. medinalis* adults fed only water is significantly lower than that of females provided supplementary nutrition [38]. However, these studies were restricted to the effect of continuous starvation on reproduction in adult *C. medinalis*. The effects of starvation at different ages and for different durations on reproduction and migratory propensity are still unclear. Most studies have used only the durations of preoviposition period to indirectly reflect migratory propensity, but little is known about effects of adult nutrition or starvation in the adult stage on take-off behavior, flight capability and energy reserve and transfer. This raises important questions about the effects of adult nutrition on the migration and reproduction of newly emerged *C. medinalis*: (1) Does starvation at different ages and for different durations have different effects on female reproduction? (2) Does *C. medinalis* (both female and male) have a critical period of response to starvation? If yes, how does starvation during this period affect reproduction and migration decisions? (3) How does adult nutrition affect energy distribution in both female and male moths?

To address these questions, we designed four independent experiments based on ecological and physiological aspects to study the effects of starvation of different durations or different timing (starvation at different starting ages) on the reproductive parameters (female only), migratory parameters and energy allocation of newly emerged *C. medinalis* and to identify the stage at which adult insects are sensitive to a lack of nectar. The results will help us understand the relationships between supplemental nutrients, migratory behavior and reproduction and elucidate the role of nectar resources in insect migration.

## 2. Materials and Methods

### 2.1. Insects

The larvae of *C. medinalis* were originally collected from rice fields in Nanjing, China. Larvae were reared on wheat seedlings until pupation [39]. Pupae were removed from the seedlings and transferred into a transparent plastic box (16 cm length, 24 cm width and 22 cm height), the bottom of which was filled with moist cotton wool to maintain high relative humidity. Pairs of newly emerged male and female adults were transferred to 500-mL transparent cups with absorbent cotton wool soaked in 5% honey solution as a supplemental nutrient. The cups were covered with plastic film, and the adults oviposited on the film. All the insects were reared in RXZ intelligent artificial climate chambers (Ningbo Jiangnan Instrument Factory, China) at 26 ± 0.5 °C and 80%–90% relative humidity (RH) with a photoperiod of 14L:10D [39]. These insects were maintained for twenty generations in the laboratory when the experiments started.

### 2.2. Adult Starvation

To explore the effects of supplemental nutrition on the reproductive and migratory ability of *C. medinalis* and to identify the sensitive period of adult response to starvation, we starved adults at different ages and for different durations. We had three treatment groups: (1) no starvation, (2) early starvation treatments and (3) no feeding (Scheme 1). Among them, the early treatment feeding group included six treatment subgroups based on different day-age durations (one, two and three days) and different starting ages (one-, two- and three-day-old). Treatments were imposed identically on male and female moths.

### 2.3. Reproductive Parameters of Female Moths

Reproductive parameters, including mating percentage, total number of eggs per female and preoviposition period were measured in treated females. After adult emergence, each female was paired with a male and transferred to a 500-mL transparent cup set up as described above. The number of eggs per female was recorded daily to calculate the preoviposition period and fecundity. After death, the female moths were dissected, and the number of spermatophores was determined; female with spermatophores copulated successfully, and moths without spermatophores failed to copulate. The mating percentage of adult females in the starvation treatments and controls were computed. More than 14 adult females were examined for each treatment.

### 2.4. Migration Parameters

Migration parameters, including take-off behavior and flight capability, were used to indicate the migration propensity of adults. Because tests of reproductive characteristics obtained sensitivity to starvation at 1 day after emergence, three treatment groups were set up to test migration characteristics: no starvation group, first-day starvation group, and adult life-long starvation group. Radar observations and field trials have shown that *C. medinalis* adults take off at dusk, approximately 19:00 [31]; therefore, observation of take-off behavior and measurement of flight capability in one-, two-, and three-day-old adults started at 19:00.

#### 2.4.1. Percent Take-Off

Adults for observation were transferred into 500-mL clear plastic cups. The cups were placed on white plastic foam (the take-off platform) and then covered with a transparent Polyvinyl chloride (PVC) cage (diameter 60 cm, height 120 cm) to observe the take-off behavior of the moths. Each time five female or male moths were observed, and at least 30 female or male moths were randomly selected for each treatment. An effective migratory take-off was defined as that moth took off and spiraled vertically with a vertical distance greater than 100 cm. If moths kept still or just flew at lower height less than 100 cm, the these moths were recorded as non-migratory [37]. Here, the number of adults that exhibited effective migratory take-off was recorded, and then take-off frequency was computed. All take-off experiments were started at dusk.

All take-off experiments were performed in a climate chamber (26 ± 1 °C, 80%–90% RH). All tested adults were moved into this climate chamber one hour before the observation of take-off behavior [37]. To simulate the lighting conditions of natural sunset, a light source composed of 20 rows of fluorescent lamps (36 V/40 W) and 2 incandescent lamps (12 V/40 W) was used. This light source was located 200 cm above the take-off platform to eliminate the effect of its heat on the internal temperature of the PVC cage. The light intensity was changed by gradually extinguishing 20 parallel fluorescent lamps (2 every 3 min) and connecting the incandescent lamp with a potentiometer to create artificially simulated evening light. The indoor light intensity was gradually decreased from 1000 L× to 0.1 L× within 45 min. The changes in light intensity during the observation period were simultaneously measured with a TES-1330A illuminometer.

#### 2.4.2. Flight Capability Measurement

Flight tests of adults were conducted with a 24-channel computer-interfaced flight mill system (Jiaduo Science, Industry and Trade Co., Ltd., Hebi City, Henan Province, China) that automatically recorded total flight distance and flight duration. Each adult was tethered according to a technique described in previous studies [40,41,42]. Experimental moths were mildly anesthetized with ether before being glued onto a tether arm. The scales of the adults at the junction between the metathorax and abdomen were brushed off using a soft brush pen, and the metathorax of each adult was glued onto a hollow plastic tether (diameter of 0.80 cm and length of 1.5 cm) with Pattex Superglue (Henkel Adhesive Co., Ltd., Guangdong, China). The flight capabilities of the adults were not affected by this treatment compared to those of the adults that were not anesthetized and glued to tethers [40,41,42]. The flight directions of the tethered moths remained perpendicular to the arm of the round-about flight mill. Light intensity was gradually reduced from 1000 L× to 0.1 L× to simulate natural dusk. Lights were turned off at 20:00 and turned on at 6:00 [37]. The flight environment was maintained at 26 ± 1 °C and 80%–90% RH, and the flight test was carried out from 19:00 to 7:00. At least 22 female or male moths in each treatment were randomly selected to observe take-off behavior.

### 2.5. Determination of Triglyceride Content

The triglyceride content in the thorax and abdomen was measured when adults were starved on the first day after emergence, starved for the life of the adult, or fed honey throughout the adult period. After treatment, one-, two-, three-, four- and five-day-old adults were collected and maintained in liquid nitrogen for determination of triglyceride content using a Triglyceride Assay Kit (Nanjing Jiancheng Bioengineering Institute, Nanjing, China) according to the manufacturer’s instructions. Each sample contained the thorax or abdomen of three moths, and at least 27 female moths or male moths in each treatment were used to determine triglyceride content. The thorax of each adult was separated from the abdomen after removal of the head, wings, and appendages. The weights of the thorax and abdomen were measured by an XP6 electronic balance (0.001 mg, Mettler-Toledo AG, Sonnenberg Strasse, Schwerzenbach, Switzerland). A proportionate amount (M:V = 1:9) of PBS (0.1 M, pH 7.4) was added, and the mixture was homogenized in ice water. The sample was centrifuged at 587× *g* for 10 min at room temperature (26 ± 1 °C), and supernatant was used for triglyceride measurement. Measurement was performed for a blank tube with 7.5 μL of distilled water, a standard tube with 7.5 μL of reference material, and a sample tube with 7.5 μL of sample homogenate. The OD value of 200 μL of the solution was measured at 510 nm using a VersaMax microplate reader (Molecular Devices) after the addition of 750 μL of working fluid. A portion of the sample supernatant was used to determine protein content using a Pierce^®^ BCA Protein Assay Kit (Thermo Fisher Scientific Inc., Waltham, MA, USA). Finally, the triglyceride levels were determined using protein levels as a quantitative standard. The formula used was as follows:(1)$$\begin{array}{ll} \mathrm{Triglyceride}\;\mathrm{content}\; & \left(\mathrm{mmol}/\mathrm{gprot}\right)\\ & =\;(\mathrm{OD}\;\mathrm{value}\;\mathrm{of}\;\mathrm{sample}-\mathrm{OD}\;\mathrm{value}\;\mathrm{of}\;\mathrm{blank})/(\mathrm{OD}\;\mathrm{value}\;\mathrm{of}\;\mathrm{standard}\\ & -\;\mathrm{OD}\;\mathrm{value}\;\mathrm{of}\;\mathrm{blank}\times \mathrm{calibrator}\;\mathrm{content}\;(\mathrm{mmol}/L)\\ & ÷\;\mathrm{protein}\;\mathrm{content}\;\mathrm{of}\;\mathrm{test}\times \mathrm{sample}\;(\mathrm{gprot}/L) \end{array}$$
where mmol/gprot is millimol per gram of protein, mmol/L is millimol per liter and gprot/L is grams of protein per liter.

### 2.6. Data Analysis

All data obtained from the studies are prented as the mean ± SE, except mating percentage and take-off percentage. The effects of adult starvation on the preoviposition period, total fecundity, triglyceride content and flight capability were analyzed by one-way ANOVA followed by Tukey’s HSD tests. The reproduction traits were also analyzed using two-way ANOVA with duration and timing of starvation. Three-way ANOVA was used to analyze the effects of different treatment on flight duration, flight distance and triglyceride content in *C. medinalis.* Differences in mating percentage and take-off percentage among treatments were compared by the chi-square test. All statistical analyses were performed with R software (version 3.6, Microsoft Corporation, Redmond, WA, USA).

## 3. Results

### 3.1. Influence of Adult Starvation on Reproduction

The control group of *C. medinalis* females did not have a starvation experience in our experiment, and its mating percentage was 80.49% (33/41). In the group of starvation throughout adult life, only 16.07% (9/56) of females mated, and this mating percentage was significantly lower than that of the control group (Chi-square test: χ^2^ = 14.09, *p* < 0.001; Figure 1a). In the groups of individuals starved for a short period (one, two or three days), mating percentages were 82.73% (115/141), 79.07% (34/43) and 71.43% (10/14), respectively. Compared with the control group, the result of Chi-square tests (χ^2^ = 0, *p* = 1 in all three comparisons; Figure 1a) indicated that a short period of starvation before oviposition did not significantly affect percentage of copulation in *C. medinalis*.

Starvation reduced female fecundity of *C. medinalis*, but the effect of starvation on the total number of eggs per female produced was found to be depended only on starvation duration, but not starvation timing (age of moth when starvation began) (Table 1, Figure 1b). The number of eggs per female decreased as starvation duration increased (Figure 1b). The female control group without starvation laid 370 ± 18 eggs (*n* = 33), while the group that starved throughout the whole adult life laid only 48.67 ± 7.84 eggs per female (*n* = 9) (Figure 1b). However, one days’ starvation (no. of eggs per female = 316 ± 12, *n* = 113) or a starvation in their first two days after emergence (no. of eggs per female = 275 ± 23, *n* = 15) did not significantly affect female fecundity compared with the control group (Figure 1b).

Starvation extended the preoviposition period of *C. medinalis* (Figure 1c, Table 1), but was affected by the starvation timing not the starvation duration (Table 1). Adults suffering starvation from the first day after emergence exhibited a significantly longer preoviposition period (4.54 ± 0.14 days, *n* = 67) than the control group (3.3 ± 0.09 days, *n* = 33) (Figure 1c). In the groups of adults starved from age 2 or 3 days, their preoviposition periods (2-day-old group: 3.9 ± 0.13 days, *n* = 63; 3-day-old group: 3.6 ± 0.11 days, *n* = 36) were not significantly different from that of the control group (Figure 1c).

Taking the above results together, we found that starvation had a negative effect on female *C. medinalis* reproduction, but a short period of starvation before oviposition did not significantly reduced the mating percent (Figure 1a) or the number of eggs per female (Figure 1b). The preoviposition period was extended significantly if moth suffered starvation in the first day after emergence (Figure 1c).

### 3.2. Influence of Adult Starvation on Flight Performance

Starvation was expected to stimulate *C. medinalis* adults to take off and start the migration process, as the preovipostion period was extended under starvation in our experiment (see previous section). As expected, the groups with a starvation experience had a higher take-off percentage, regardless of sex (Table 2, Figure 2a,b). In the control group without starvation experience, only 9.40% (17/180) individuals took off in the first 3 days after emergence (Table 2, Figure 2b,c). All groups experiencing starvation at the age of 1 day had a low take-off percentage, only 14.29% (20/140), which was similar to the control group (Chi-square test: χ^2^ = 1.70, *p* = 0.19). This indicated that most *C. medinalis* moths did not commence their migration immediately after emergence. The takeoff percentages in the groups starved on day 1 increased from the second day of age (Figure 1b,c). The group starved for the first three days of adult life had a strong migratory propensity at 2- and 3-days old (takeoff percentage at day 2: 67.14% (47/70); day 3: 92.86% (65/75)). In the group starved on day 1, takeoff percentages were 31.43% (22/70) o the second day, and 48.57% (34/70) on the third day. These takeoff percentages were higher than that of control group (Chi-square test, day 2: χ^2^ = 5.94, *p* = 0.015; day 3: χ^2^ = 7.56, *p* = 0.006), but lower than that of the group starved for 3 days (Chi-square test, day 2: χ^2^ = 5.44, *p* = 0.019; day 3: χ^2^ = 4.12, *p* = 0.042) (Figure 2a,b). Here, it should be emphasized that the takeoff percentage in the groups with one-day’s starvation were higher at the age of 3 days than at the age of 2 days (Chi-square test: χ^2^ = 37.47, *p* < 0.001). This result indicates that feeding after starvation did not reduce migration propensity even if the decision to migrate was initiated by starvation.

In the control group of *C. medinalis* without starvation, a short preoviposition period and a low takeoff percentage indicated that it did not have a strong propensity to migrate (see previous sections), but it still had strong flight capability before oviposition (Figure 2b,c). The result of ANOVA indicated that flight duration was significantly different between female and male individuals (F: 4.84 ± 0.43 h (*n* = 72); M: 3.31 ± 0.33 h (*n* = 88); *F*_1,156_ = 9.71, *p* = 0.002), but just a marginally significant difference between individuals at different ages (*F*_2,156_ = 2.94, *p* = 0.056) (Figure 2c,d). In the group starved for duration of adult life, the flight duration in the first 3 days after emergence was 2.08 ± 0.24 h (*n* = 78), and there was not significantly difference between female and male individuals (*F*_1,74_ = 1.16, *p* = 0.29), or individuals at different ages (*F*_1,74_ = 1.18, *p* = 0.31). Nonetheless, we found that the flight capability of this group decreased with age (but not significantly), while the flight capability of the control group increased (marginally significant). Consequently, the flight capabilities of these two groups were significantly different at age 3 days (Figure 2c,d). This result indicated that starvation reduced the flight capability. In the group of individuals with one-day’s starvation at age 1 day, flight duration in the first 2 days was 2.30 ± 0.29 h (*n* = 84), regardless of sex (*F*_1,74_ = 1.15, *p* = 0.29) or age (*F*_1,81_ = 0.31, *p* = 0.58). Also, this flight capability was similar to that of other two groups (i.e., control and adults starved for life) at the same age (Figure 2c,d). At an age of 3 days, the flight duration of this group after one-day’s starvation increased to 5.70 ± 0.52 h (*n* = 54), and this was even longer than that of the control group at the same age (just below significant). The result of flight distance was just similar to that of flight duration (Figure 2e,f), so the details of flight distance are not shown here.

In summary, starvation stimulated *C. medinalis* to fly and begin migration. The takeoff percentage did not decrease even if supplemental nutrient was supplied by feeding with honey solution (Figure 2a,b). The flight capability remained at a nearly stable level at the first 3 days after emergence even under starvation. Supplemental nutrition enhanced their flight capability, especially in the individuals starved for 1 day (Figure 2c–f). Thus far, the results suggest that migratory moths allocate new energy to migration but not to reproduction.

### 3.3. Influence of Adult Starvation on Triglyceride Content

The triglyceride content in the thorax of *C. medinalis* was not significantly different among individuals of different sexes, ages or starvation treatments in our experiment (see more statistics detail in Table 3; Figure 3a,b), and was expected to help moths maintain a nearly stable flight capability during their first three days after emergence. However, the abdominal triglyceride content varied depending on treatment and an interaction effect between treatment and age (Table 3). In the control group, the abdominal triglyceride content increased with age, suggesting that moths allocate new energy to reproduction (Figure 3a,b). In the group starved throughout the whole adult period, the abdominal triglyceride content decreased significantly at age 3 days (Figure 3a,b). As flight capability did not decrease significantly and the triglyceride content in the thorax remained unchanged, it appeared that moths transferred energy from reproductive tissues to flight tissues to maintain migration capability. The group starved only on day 1 after emergence had a stable abdominal triglyceride content.

## 4. Discussion

### 4.1. Effect of Adult Starvation on Reproductive Parameters

*C. medinalis* moths have the ability to endure short-term starvation because of energy stored during the larval stage, but reproduction of the adult moths is strongly dependent on supplemental nutrition. Mating is costly for both sexes and increases energy expenditure [43]. Long-term starvation results in failure to mate adequately, which greatly affects the reproduction of insects. Many lepidopteran insects exhibit greatly reduced mating percentages, such as *Spodoptera exigua* [22] and *Pseudaletia separata* [21], when adult moths lack nutrition. In addition, even if adults successfully mate, females tend to reduce oviposition to adapt to starvation [44,45,46]. Most lepidopteran insects, such as *Spodoptera exempta* [47], and *Athetis lepigone* [48], show significant decline in fecundity when adults are undernourished. In this study, the fecundity of *C. medinalis* was noticeably reduced when starvation lasted for more than two days. Fecundity and ovarian development levels of *C. medinalis* females fed only with water were significantly lower than those of adults provided supplemental nutrition [12,38]. Therefore, adult nutrition is particularly important for reproduction and lack of nectar may strongly impact reproduction success in *C. medinalis*.

In general, to escape starvation, insects have evolved a series of strategies to deal with food shortages. These include physiological factors, such as reducing metabolic rate to save energy [49] to deal with starvation and behavioral responses, such as migration [1], to avoid starvation. As an insect with long-distance migration ability, *C. medinalis* can escape from bad environments and track resources by migration through time and space. In a previous study in southern China, it was found that *C. medinalis* emigrants (old instar larvae or pupae were collected from rice paddies during the emigration period of natural population and then fed indoor) had much lower fecundity (laid 84.76 ± 59.74 eggs per female) than residents (339.47 ± 115.36 eggs per female), but emigrants remained stronger migration propensity through days one to six after emergence [37]. By contrary, immigrants (completed its migration, moths collected from natural population during its immigration period and then fed indoor) laid the biggest number of eggs (400.50 ± 70.07 eggs per female) [37,50]. It seems that migratory females reduce reproductive investment to support migration process until the termination of migration, and migrants allocate most resource to reproductive after migration and even had a higher fecundity than residents [37,50]. Therefore, we speculate that *C. medinalis* might need to reduce reproductive investment after short-term starvation in favor of allocating resources to searching for a more resource-rich environment.

### 4.2. Effect of Adult Nutrition on Behavioral Response

Adult starvation affects the timing of oviposition in *C. medinalis*, that is, the preoviposition period. Migration of *C. medinalis* begins while the adult is sexually immature, and ovarian development of females undertaking migration is slowed [32], resulting in longer preoviposition periods in migratory populations [37]. Therefore, prolongation of the preoviposition period is closely related to insect migration. Our results indicate that starvation significantly prolonged the preoviposition period of *C. medinalis*, which is consistent with the results of Zhang et al. [12]. In addition, there is a crucial stage in which some migratory insects respond to environmental factors, and during this crucial period, environmental changes can alter the direction of insect development, such as by inducing migration or reproduction [51,52,53]. Our study found an interesting phenomenon where starvation on the first day post-eclosion prolonged the preoviposition period. Other studies in *C. medinalis* also found that only the first-day adults suffering low temperature or treated by juvenile hormone shorten the preoviposition period [41,54]. Therefore, we believe that the first day after emergence is a critical period for *C. medinalis* to respond to environmental changes, and that starvation on that day promotes the onset of migration.

Our study showed that starvation in recently-emerged adults increased take-off percentage, and the longer the starvation duration, the higher the take-off percentage. Thus, adult starvation enhances propensity to migrate. Interestingly, the take-off percentage of *C. medinalis* subjected to restoration of nutrition after one day of starvation did not decrease but was significantly higher than that of the control group on the third day of adulthood, which indicated that starvation on the first day greatly enhanced migratory propensity. This result confirms that the first day after emergence is a critical period for *C. medinalis* to respond to environmental changes.

In our study, only starvation for more than two days decreased the flight capability. Such short-term starvation does not reduce flight capability in some insects but is exhibited by others, such as *Pachnoda siuata* [55] and *Agrotis ipsilon* [56]. Although the flight capability of *C. medinalis* measured by tethering flight mills did not show significant differences between adults in the short-term starvation groups and control, starvation might reduce the flight capability in some other non-migrating insects [57]. This may be due to the fact that *C. medinalis* deals with starvation differently from other migrating insects. The accumulation of energy during the larval period is limited, and when adults face starvation, energy may need to be redistributed from reproduction organs to flight organs, or reversely. As a migratory insect, *C. medinalis* are more inclined to allocate energy into flight organs to ensure that there is sufficient flight capacity to emigrate away from a food-poor environment, while other insects may reduce energy allocation to other organs to ensure reproduction. However, in other migrating insects such as *Spodoptera exigua* [58], starvation has been found to reduce flight capacity. This difference may be due to a difference in energy requirements between migratory insects. There was no significant change in flight capability in the case of days one to three after emergence of *C. medinalis*, but the flight capability of *S. exigua* increased with adult age [59]. When energy is limiting, *C. medinalis* can maintain flight capacity by redistributing triglyceride from the abdomen to thorax, but the flight of *S. exigua* requires more energy, and the storage of energy during larval stage is extremely limited [59]. However, the specific impact mechanism needs further research. Therefore, *C. medinalis* apparently has a strategy to allocate energy towards flight when it encounters a food shortage early in its adult life.

Lepidoptera adult behavioral strategies are directly or indirectly altered by changes in adult nutrition. Food shortage in newly emerged *C. medinalis* prolonged the preoviposition period, increased take-off propensity and maintained the long-distance flight capability. These behaviors showed a strong migratory propensity in *C. medinalis* and are consistent with our previous predictions, based on reproductive characteristics, that starvation induces migration.

### 4.3. Effect of Adult Nutrition on Physiological Adjustment

Our experiments clearly demonstrated that in the face of starvation, *C. medinalis* has prolongs reproduction development and increases migration propensity. Behavior results from complex physiological processes, and energy redistribution is one of the adaptive physiological countermeasures to food shortage. The transition between migration and reproduction results from the transformation or redistribution of intrinsic energy [37,55]. Because *C. medinalis* exhibits “oogenesis-flight syndrome” during its long-distance migration [33,34,50], there is a trade-off between flight and reproduction [9]. Moreover, triglycerides are the main energy sources for migration of *C. medinalis* [9,35,36]. Consequently, the triglyceride distribution in the thorax and abdomen reflects the transition between migration and reproduction. In this study, we found that, regardless of sex of the moth, its thoracic triglyceride content is not significantly reduced by starvation, indicating that *C. medinalis* maintains the energy supply to the thorax under hunger. In contrast, starvation affects the energy reserve in the abdomen. Our results show that when nutrition is supplemented, the triglyceride content of both female and male moths increases with the adult age, while adults who continue to be deprived have reduced triglycerides. However, moths starved on the first day did not quickly replenish their abdominal triglycerides after supplementing nutrition, which may be related to their prolonged preoviposition period and higher takeoff tendency on the third day after emergence. These results suggest that *C. medinalis* chooses between breeding and migration in response to nutritional deprivation. When food is abundant, *C. medinalis* choose to allocate more energy into breeding, while faced with starvation, it will allocate more energy into the flight-readiness. Somewhat surprisingly, although continuous starvation did not significantly reduce the triglyceride content of the thorax, the flight capability decreased on the third day after emergence. Since starvation can lead to a decrease in metabolic enzyme activity [55,60,61,62], we speculate that this may be related to the decline in activity of flight-related enzymes under starvation that affect energy efficiency, but the specific reasons are not clear and further research is needed to validate our speculation. In short, changes in physiological adjustment after starvation explained the results of reproductive characteristics and behavioral response. Starvation in the first day of *C. medinalis* life leads to the allocation of energy into their thorax, ensuring successful migration.

## 5. Conclusions

Overall, our study shows that a short starvation (one or two days) before preoviposition don’t significantly reduce the fertility and flight capability of *C. medinalis*, and adult supplemental nutrition is essential for the reproduction of *C. medinalis* moths. One day after emergence is a critical period in the response to starvation. Starvation during the critical period promotes migratory traits and leads to migration in *C. medinalis*. Our results suggest that nectar resources play an important role in insect migration, and may be important for development of improved ecological control strategies.

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
