# Peer review of "Cnaphalocrocis medinalis Moths Decide to Migrate when Suffering Nutrient Shortage on the First Day after Emergence"

_insects, 2019, doi:10.3390/insects10100364_

Round 1

Reviewer 1 Report

This study looks at the effects of nutrition on newly eclosed adult moths, particularly the effects of varying periods of starvation on physiology and behavior related to migration.

They document the importance of the first adult day, and the various ramifications of fasting on migration. In general the methods seemed appropriate and well described, although I do not have experience with triglyceride analysis or flight mills. The current manuscript would benefit from light editing for English language syntax and style.

On line 64, the authors suggest that the suite of migratory adaptations includes maintaining sufficient energy reserves. Is there any information about what amounts of triglyceride stores are needed to undertake migratory flight? Does this energy come solely from the stored energy in the thorax, as measured in this study? More background information would be helpful, if available.

Other minor comments:

Table 1 – treatment schemes

I’m assuming adult age numbers refer to days

Line 223 – starvation extended preoviposition period more from duration than timing, according to Table 2.

Line 251 – I recommend not using the word ‘appetite’ here, because it can be confusing when part of the experiment involved starvation.

Line 329 – I don’t understand this sentence (begins with “The time”). In this study there were different amounts of time studied.

Author Response

The current manuscript would benefit from light editing for English language syntax and style. >>> The English writing was improved by Reviewer #2’s carefully editing. On line 64, the authors suggest that the suite of migratory adaptations includes maintaining sufficient energy reserves. …. More background information would be helpful, if available. >>> Added. “C. medinalis migrants also reserve sufficient energy for migration [35,36]. Lipids, stored primarily as triglycerides, are the main energy sources for migration in many nocturnal moths [9], and C. medinalis migrants have more lipids stored in thorax than residents [35,36].” Table 1 – treatment schemes I’m assuming adult age numbers refer to days >>> ‘Adult age’ was changed to ‘Adult age (day)’ Line 223 – starvation extended preoviposition period more from duration than timing, according to Table 2. >>> Added Line 251 – I recommend not using the word ‘appetite’ here, because it can be confusing when part of the experiment involved starvation. >>> ‘appetite’ was replaced with ‘propensity’ Line 329 – I don’t understand this sentence (begins with “The time”). In this study there were different amounts of time studied. >>> Agree, it is not a clear statement, and mislead readers. Also, this sentence did not any important information. So, it was deleted.

Reviewer 2 Report

The paper has merit for publication from the perspectives of both agriculture and ecology. It shows that there is a critical period in the life of an adult rice leaf roller when it "decides" to migrate from the larval area, based on availability of sugary adult food directly after the adult emerges, and it provides physiological evidence for the mechanism behind the behaviors. This has implications for pest management through habitat manipulation.

The authors are commended for the great effort they made to write the paper in English. The text is mostly understandable but it needs small improvements in many places, and moderate improvements in a few places. To save time, I made changes directly in the pdf and highlighted the changes in yellow.  The comments and questions are also placed directly on the pdf.

Several of my changes were "best guesses", so the authors must review them very carefully to ensure that they are correct. Also, please provide more background information about the moth's biology and briefly describe the role of triglycerides in moth or insect energetics. I am not familiar with C. medinalis (although I am familiar with many other moths), and it has been many years since I have read about the induction of migration in insects. I suspect that other readers will benefit from this information.

Author Response

In the introduction, please provide more information on the biology of the species (see note below); also, briefly describe the role of triglycerides in insects.

>>>  Added.

Biology information: “Cnaphalocrocis medinalis (Lepidoptera: Pyralidae) is an important long-distance migratory pest, that has had serious outbreaks in many Asian countries in recent decades, especially in China [31,32]. It cannot overwinter in temperate climates. In China, only a few individuals can survive during winter in areas of Southern Yunnan, Southern Guangxi, Southern Guangdong and Hainan [31-33]. Instead, infestations in temperate zones are initiated by windborne spring/summer migrants from the southern areas. A northward migration begins in March every year, and migrant progeny further expand population distributions to cover the rice-growing regions in China, Japan and the Korean Peninsula [31-34]. From September onwards, the general direction of C. medinalis becomes predominantly south bound [31-34].”

“Like most nocturnal migratory moths, C. medinalis perform ‘multi-stop’ migration in 3-5 consecutive nights, in which moths only take off at dusk, terminate migratory flight the following dawn, and then take-off again at the next dusk[33]."

The role of triglycerides:C. medinalis migrants also reserve sufficient energy for migration [35,36]. Lipids, stored primarily as triglycerides, are the main energy sources for migration in many nocturnal moths [9], and C. medinalis migrants have more lipids stored in thorax than residents [35,36].”

L53 “However, these studies often determine migratory propensity only by field observation and changes in reproductive characteristics.”

Not clear: Changes in which reproductive characteristics?

>>> I agree, and this sentence did not any important information. So, it was deleted.

Please explain if there is a difference between the sexes, too.

Please clarify: do these questions apply to females, or to both sexes?

>>> The reproductive parameter was only applied to female, and other parameters were to both sexes.  I already made some change to clarity it.

Diffusion should be defined in the introduction, not here.

Please define "spiral vertical".  Is it the vertical distance traveled or the full distance along the spiral movement?  How can you measure the spiral distance?

>>> ‘Diffusion’ was removed here. Whole sentence was rewritten.

“An effective migratory take-off was defined as that moth took off and spiraled vertically with a vertical distance greater than 100 cm [9]. If moth kept still or just flew at lower height less than 100 cm, this moth was recorded as non-migratory [9].”

Device?  Earlier you described it as a white plastic foam platform.

>>> All experiments were performed in a climate charmer, a light source was used to simulate the light condition of nature sunset. A PVC cage used to observe the take-off behavior. To make clearer, we rewrote the part of “Percent take-off”.

All "gender" in this paper was replaced with "sex".

>>> Thanks very much.  All ‘gender’ words were changed to ‘sex’.

L329: The time to endure starvation after emergence in C. medinalis is very short, only one day.

I cannot understand this sentence.  Please rewrite it.  Why is the time to endure starvation only one day?  Does the moth live only one day in the field without food?

>>> Agree, it is not a clear statement, and mislead readers. Also, this sentence did not any important information. So, it was deleted.

Please explain how mating percentage in emigrants vs residents is important to your argument about energy trade-offs. 

The original statement was contrary to most evolutionary thought.  The moths are not ensuring the continuation of the population, they are ensuring their own fitness.

Please consider the change carefully.

>>> These sentences were rewritten.  In Zhang et al., 1979, the mating percentage was very low in emigrants, about 28%, while > 90% in the residents and immigrants.  But this result cannot give any useful evidences to support our statement. Because in his paper, moths were collected from field directly. The emigrant moths were dissected before migration, and the age were younger than residents and immigrants. So I removed the reference. In Yang et al., 2013, the reproductive parameters were also compared between migrants and residents. She collected old instar or pupae from natural population, and fed them indoor, and found there are significant difference between migrants and residents through their life.  I think Yang et al., 2013 were more creditable. 

L351: ‘Adult starvation can prolong the reproductive preparation time in C. medinalis, which is conducive to the onset of migration.’

Is starvation conducive to onset of migration, or is reproductive readiness negatively correlated with starvation?

>>>  There was not strong evidence to make this conclusion, so I deleted this sentence.

L356: Please explain: How are they sensitive to temperature?  How is this sensitivity related to your conclusions?

>>> The sentence was changed to “Other studies in C. medinalis also found that only the first-day adults suffering low temperature or treated by juvenile hormone shorten the preoviposition period [41,54].”  

This statement can be confused by some readers because it looks like circular logic. It is a good idea to explain that all moths can take off, but that moths that migrate take off more frequently than residents. 

>>> Agree. The true means is that individuals from migratory population take off more frequently than that from a resident population. Anyway, the first two sentence in this paragraph mislead readers, and did not give some important information. So I deleted this sentence.
